# Daily Fluctuation of Facial Pore Area, Roughness and Redness among Young Japanese Women; Beneficial Effects of *Galactomyces* Ferment Filtrate Containing Antioxidative Skin Care Formula

**DOI:** 10.3390/jcm10112502

**Published:** 2021-06-05

**Authors:** Kukizo Miyamoto, Bandara Dissanayake, Tatsuya Omotezako, Masaki Takemura, Gaku Tsuji, Masutaka Furue

**Affiliations:** 1Research and Development, Kobe Innovation Center, Procter and Gamble Innovation GK, Kobe 651-0088, Japan; omotezako.t.1@pg.com; 2Research and Development, Beauty Care, P&G International Operations (SA) Singapore Branch, Singapore 138547, Singapore; dissanayake.b@pg.com; 3Department of Dermatology, Graduate School of Medical Sciences, Kyushu University, Fukuoka 812-8582, Japan; take0917@dermatol.med.kyushu-u.ac.jp (M.T.); gakku@dermatol.med.kyushu-u.ac.jp (G.T.); furuemasutaka00@yahoo.co.jp (M.F.); 4Research and Clinical Center for Yusho and Dioxin, Kyushu University Hospital, Fukuoka 812-8582, Japan

**Keywords:** facial imaging, young women, pore area, roughness, redness, fluctuation, *Galactomyces* ferment filtrate

## Abstract

Young women often complain about the daily fluctuation of their facial skin conditions. However, no objective study has been carried out on such changes. This study is aimed at quantitatively elucidating daily skin fluctuation and evaluating the efficacy of cosmetic skin care treatment. We developed the first portable and self-guided facial skin imaging device (eMR Pro) to reproducibly capture facial images at home. Two 8 week clinical studies were then conducted to analyze daily skin fluctuation of facial pore areas, roughness and redness in young Japanese women (*n* = 47 in study 1 and *n* = 57 in study 2) by collecting facial images three times a day, during the morning after wake-up, during the morning after face wash, and during the evening after face wash. After a 4 week baseline measurement period (week -4 to week -1), all subjects applied *Galactomyces* ferment filtrate (GFF, Pitera^®^) skin care formula twice a day for 4 weeks (week 1 to week 4). These three skin conditions did exhibit different fluctuation patterns. The pore area and roughness showed the “morning after wake-up”-largest fluctuation pattern, whereas redness showed the “evening after face wash”-largest fluctuation pattern. GFF treatment significantly reduced the net values and delta fluctuation of pore area, roughness, and redness, which were consistently observed in two studies. In conclusion, the daily fluctuation of facial skin conditions is potentially a new target field for investigating healthy skin maintenance.

## 1. Introduction

It is important to maintain the homeostasis of epidermal bioavailability and structural functions for healthy and stress-tolerant skin with a youthful appearance. Various studies on the appearance of skin aging have been carried out by measuring chorological aging among multiple ethnicities [1,2,3,4,5]. We also reported that skin moisturization is critical for a youthful appearance in a 10 year longitudinal skin aging study in Akita, Japan [6].

Our institutional customer service questionnaire survey enrolling 250,995 women aged 20–35 years demonstrated that the top three facial concerns are enlarged follicular pores, dryness/rough texture, and facial color (redness and spots). These young adult women also complain that their facial condition fluctuates daily or weekly in their usual life. However, there are no studies that address the variation or fluctuation of facial skin appearance.

Measuring the fluctuation of skin condition may be difficult in young women because their facial skin exhibits much fewer visible signs of aging compared to older people. Therefore, only questionnaire-based and subjective-type methods are applicable to evaluate skin concerns such as sensitive or irritable skin in the young population [7].

In this study, we developed a simple and self-operational but quantitative facial skin imaging system called eMR Pro. The eMR Pro system enables facial skin image collection at any time in 24 h with constant brightness and positioning. Using the eMR Pro device, two separate clinical studies were conducted under the same protocol among young Japanese women to analyze daily fluctuation of facial skin pore area, roughness, and redness. We found that facial conditions exhibit a marked daily (morning-to-evening) and different fluctuation pattern and that *Galactomyces* ferment filtrate (GFF) skin care formula (Pitera^®^, P&G Godo Kabusikigaisha, Kobe, Japan ) application can alleviate such fluctuation. These results suggest that the daily fluctuation of facial skin conditions is a potential new target field for investigating healthy skin maintenance.

## 2. Materials and Methods

### 2.1. Portable and Self-Guided Facial Imaging System: eMR Pro

The eMR Pro is a self-facial imaging system developed by P&G Company, Kobe, Japan (IP filing ref. number AA1280) and it utilizes a commercial smartphone camera/LED lighting source (iPhone 7/8/SE) (Figure 1).

By using the eMR Pro image capturing application, the subjects take pictures on the right half and left half of the face by themselves. A plastic face attachment is equipped to the smartphone to control the intensity of the LED light illuminating the skin surface (Figure 1A,B). The plastic face attachment ensures constant distance between camera head and skin surface. In order to capture consistent facial images by oneself, a small repositioning mirror and auto eye/face live-view recognition algorithm was incorporated into the eMR application program. The eMR Pro was distributed to each subject and they took their own facial images at home every day. Captured images (1080 × 1920 pixels) were encrypted and sent to the eMR Pro server (Figure 1C).

### 2.2. Facial Skin Image Analysis

Prior to the skin analysis, post color calibration of all collected eMR Pro images was performed using gray color tips mounted in the eMR Pro face attachment. The facial pore area, surface roughness, and redness image analysis algorithm were carried out using the eMR calculation software (P&G Godo Kabusikigaisha, Kobe, Japan). Each value was expressed as an arbitrary unit (au).

Briefly, pores were detected in an image analysis algorithm to detect circular pore-like shapes via edge-enhanced binary images at the cheek region. The total area of detected pores was then calculated (Figure 2A,B) [8].

Facial surface roughness was analyzed to quantify the ununiformed distribution spectrum of relative shadows caused by fine skin surface topography in the region of the eye around to cheek.

Redness was measured at the cheek by the variation and brightness of hemoglobin chromophore tone distribution by a Fourier transformation filtering the skin image (Figure 2C,D).

### 2.3. Clinical Study Protocol

Two separate 8 week clinical studies were carried out to grasp the skin appearance variation of pore area, roughness, and redness (Studies 1 and 2). In Study 1, 48 young Japanese women from age 22 to 34 (mean ± SD; 28.9 ± 4.33) were enrolled. All subjects took their facial images using eMR Pro three times a day: during the morning after wake-up, during the morning after face wash, and during the evening after face wash (Figure 3).

Subjects waited 10 min after washing their faces before taking the photos. We measured baseline data from week -4 to week -1 (Figure 3). Then, the subjects applied the GFF skin care formula from week 1 to week 4. An amount of 3 mL GFF (Pitera^®^) was applied twice daily over the face after face wash during the morning and evening. GFF (Pitera^®^) is a yeast derived extract used as a moisturizing agent in cosmetics and GFF skin care formula contains more than 90% GFF (Pitera^®^) without other skin conditioning ingredients [9].

In addition, skin hydration was measured with a corneometer reading at the cheek region at week -1, 2, and 4, respectively. Study 1 was carried out from 22 August 2018 to 4 November 2018, in Kobe, Japan.

Study 2 was conducted from 22 July 2019 to 24 September 2019, in Kobe, Japan, with the same protocol as Study 1. It included 57 young Japanese women from age 22 to 34 (mean ± SD; 29.3 ± 4.42). Data acquisition and analysis were performed in compliance with protocols approved by the Ethical Committee of Global Product Stewardship in P&G Innovation Godo Kaisya (ethical approval number CT18-001 for Study 1 and CT19-010 for Study 2, respectively). Written informed consent was obtained from all participants prior to study placement.

### 2.4. Production of Sebum from SEB1

SEB-1, immortalized human sebocytes, were cultured in standard medium consisting of DMEM with 5.5 mM glucose/Ham’s F-12 3:1, fetal bovine serum 2.5%, adenine 1.8 × 10^−4^ M, hydrocortisone 0.4 μg mL^−1^, insulin 10 ng mL^−1^, epidermal growth factor 3 ng mL^−1^, and cholera toxin 1.2 × 10^−10^ M [9]. They were treated with or without ultraviolet ray B (UVB) irradiation (25 mJ/cm^2^) in the presence or absence of 10% GFF obtained from P&G Godo Kaisya, Kobe, Japan [10]. After 24 h incubation, the sebum production of SEB-1 cells was visualized using an Adipocyte Fluorescence Staining Kit (Cosmo Bio Co., Ltd., Tokyo, Japan).

### 2.5. Statistical Analysis

SPSS 22 (IBM^®^ SPSS^®^ Statistics 26, New York, NY, USA) for Windows 2010 was used for statistical analyses. Data were analyzed by nonparametric, paired, and unpaired tests including, if appropriate, the analysis of (co-)variance and Welch’s *t*-test. The level of significance was assumed at *p* values less than 0.05.

## 3. Results

In Study 1, 47 out of 48 subjects completed the study and 7722 facial images were collected and analyzed. In Study 2, all 57 subjects completed the study protocol and 11,037 facial images were collected.

Facial pore areas showed a significant daily fluctuation. In the baseline measurement period from week -4 to week -1, pore areas were significantly larger during the morning after wake-up than those during the morning after face wash or during the evening after face wash in both Studies 1 and 2 (Figure 4 and Appendix A).

In order to better compare the morning-to-evening fluctuation in the baseline measurement period, we plotted weekly average values of the pore area (Figure 5).

The morning-to-evening daily fluctuation of pore area was clearly depicted in the baseline period from week -4 to week -1 both in study 1 (Figure 5A) and study 2 (Figure 5B). The “morning after wake-up” largest fluctuation was evidenced during each week in two independent studies. Notably, topical application of GFF reduced the pore area. When compared with baseline values at week 4, GFF treatment reduced the pore area more conspicuously during the morning after wake-up (Figure 5). As the morning-to-evening fluctuation width was also likely to diminish (Figure 5), we calculated the delta fluctuation of pore area (highest minus lowest values among three time points in a day). The delta fluctuation of pore area was also significantly alleviated after GFF treatment (Figure 6 and Appendix A).

Facial roughness showed a fluctuation pattern similar to that of pore area. In the baseline measurement period from week -4 to week -1, the roughness values were significantly larger (worse) during the morning after wake-up than those during the morning after face wash or during the evening after face wash in both Studies 1 and 2 (Figure 7 and Appendix A).

The weekly fluctuation pattern also demonstrated that the “morning after wake-up” is the largest figure in both Studies 1 and 2 (Figure 8).

In addition, GFF treatment significantly reduced the roughness values from week 1 to week 4 (Figure 8). The delta fluctuation of roughness was also significantly alleviated after GFF treatment (Figure 9).

On the other hand, facial redness showed an opposite daily fluctuation pattern. In baseline measurement period from week -4 to week -1, the redness values were significantly larger (worse) during the evening after face wash than those during the morning after wake-up or during the morning after face wash in both studies 1 and 2 (Figure 10 and Appendix A).

The weekly fluctuation also exhibited that the “evening after face wash” was the largest pattern at the baseline period in both Studies 1 and 2 (Figure 11).

In addition, GFF treatment significantly reduced the redness values from week 1 to week 4 (Figure 11). The delta fluctuation of redness was also ameliorated after GFF treatment (Figure 12 and Appendix A).

Finally, the Studies 1 and 2 were conducted at different weather situation. However, there were no significant differences in the pore area, roughness, and redness between Studies 1 and 2 (Appendix A).

## 4. Discussion

In conventional in vivo skin research, subjects are asked to visit a clinical facility weekly or monthly to collect the necessary data. We have developed a new, easy, quick, and self-guided facial imaging tool, eMR Pro, to quantitatively evaluate time-series facial skin appearances. Using this self-imaging system, we can collect facial images daily (three times a day in this study) without visits to special facilities being necessary.

We successfully quantified the facial pore area, roughness, and redness with this new imaging tool in a contactless manner among young women who usually do not show visible chronic skin aging signs, such as wrinkles and hyperpigmented spots. We analyzed the facial images acquired during the morning after wake-up, morning after face wash, and evening after face wash. Notably, we found that these three skin conditions exhibited differential morning-to-evening fluctuation patterns in two independent clinical studies.

The facial pores became the most visible or enlarged during the morning after wake-up. While the exact reasons remain unclear, there are at least two possible explanations namely, (1) the dehydration of skin surrounding pores at night and (2) the retention of sebum by head-rest position while asleep. In addition, the enlarged pore size in the morning after wake-up was significantly reduced after GFF (Pitera^®^) treatment. The delta fluctuation of pore area was also alleviated by GFF treatment.

In general, when pores and their surrounding skin are well hydrated, the stratum corneum in and around pores is swollen and the pore area is reduced. Dehydration of stratum corneum may occur at night, potentially contributing to pore area enlargement. As GFF formulated skin care product actually increased the skin hydration in this study (Appendix A), the moisturizing capacity of GFF product may be attributable to the significant decrease in pore area by GFF treatment. Decreases in pore area after face wash may also be explained by the tentative hydration by face wash. With regard to sebum production, as GFF is a potent antioxidative agonist for aryl hydrocarbon receptor (AHR) [9,11,12], it may affect sebum production. Our preliminary experiments showed that GFF significantly decreased the sebum production of sebocytes stimulated by UVB irradiation in vitro (Appendix A). Therefore, in addition to its moisturizing effect, GFF may potentially reduce the pore size by attenuating sebum production.

Skin roughness is likely to be associated with potential dehydration at night. Therefore, it also showed the “morning after wake-up”-largest fluctuation pattern similar to pore area. The skin roughness net values and its delta fluctuation were also minimized by the application of GFF.

By contrast, redness values showed the “evening after face wash”-largest fluctuation pattern. We assume that this may be attributable to a micro inflammation caused by external stresses such as casual UV exposure and environmental PM2.5 during the daytime. Daily application of GFF formulated skin care products also improved the redness values and their delta fluctuation. GFF is known to enhance the expression of skin barrier-related proteins such as filaggrin and loricrin via AHR activation [9]. GFF also inhibits oxidative stress by proinflammatory cytokine via activation of the antioxidative system [11,12]. These barrier-maintaining and antioxidative functions of GFF may be attributable to the improvement of redness.

In conclusion, as has been perceived by our customers, facial pore area, roughness, and redness showed a daily fluctuation. The pore area and roughness demonstrated the “morning after wake-up”-largest pattern, while redness exhibited the “evening after face wash”-largest pattern. The GFF-containing skin care product (Pitera^®^) reduced the net values as well as delta fluctuation of facial pore area, roughness, and redness. Further studies are warranted to clarify the mechanism and intervention of daily fluctuation of these skin conditions.

## Figures and Tables

**Figure 1 jcm-10-02502-f001:**
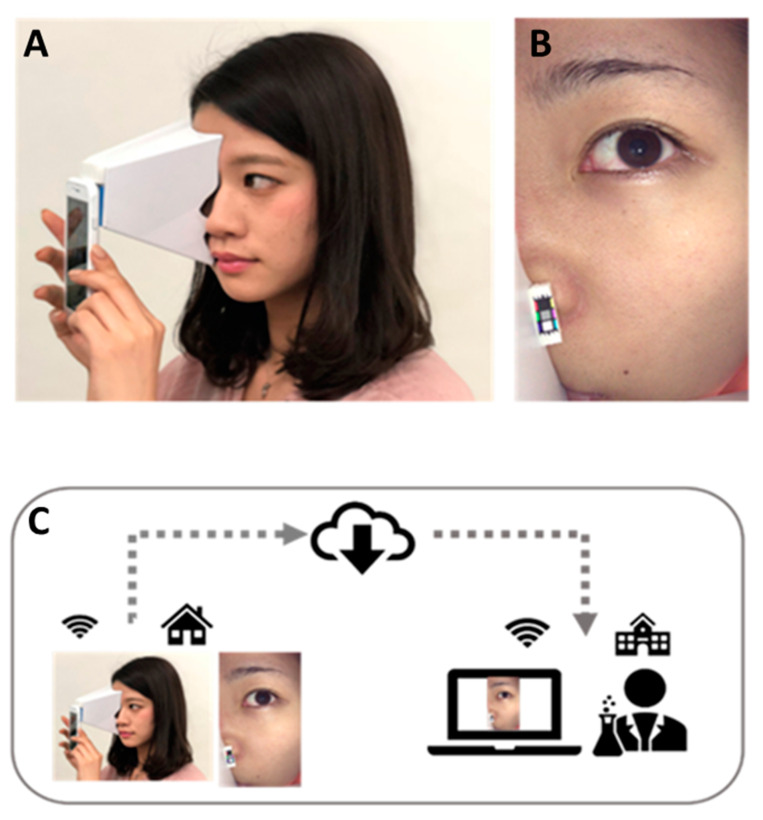
eMR Pro self-guided facial imaging system. The subjects took facial images using their own smartphones with the plastic face attachment (**A**). The facial images (**B**) were then sent to the eMR Pro server (**C**).

**Figure 2 jcm-10-02502-f002:**
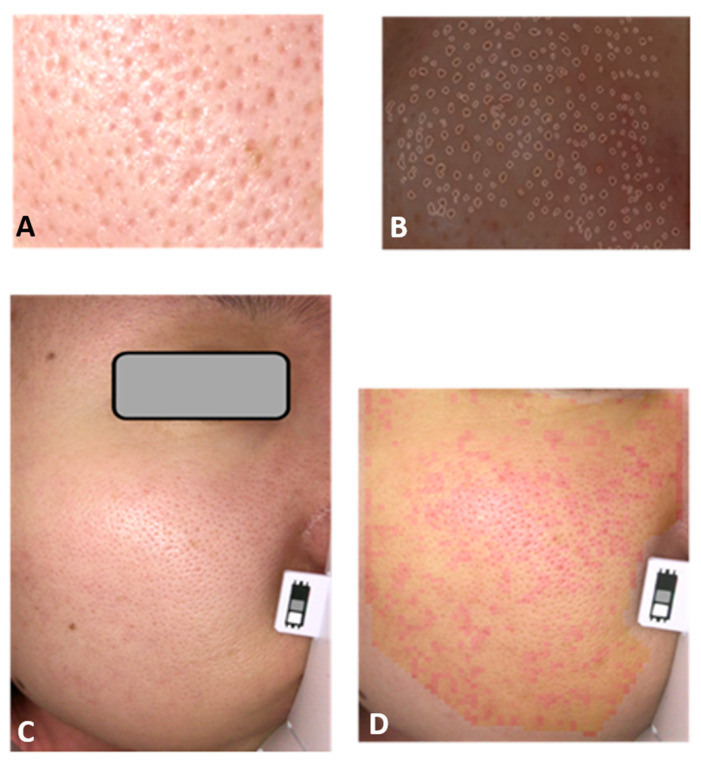
Calculation of pore area and redness variation. Facial pores at cheek (**A**) were analyzed by the image analysis algorithm to detect circular pore-like shapes via edge-enhanced binary images (**B**). Redness variation of cheek (**C**) was based on the variation and brightness of hemoglobin chromophore tone distribution by a Fourier transformation (**D**).

**Figure 3 jcm-10-02502-f003:**
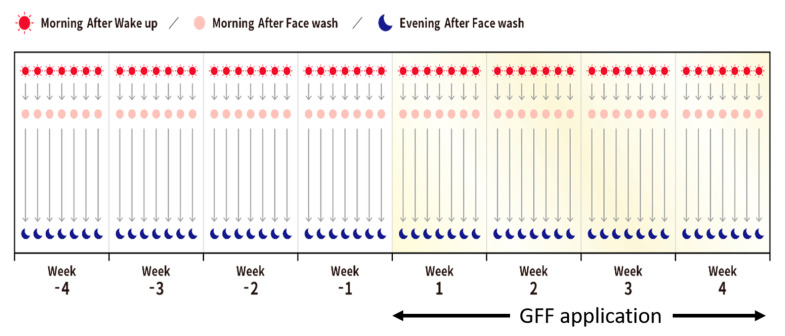
Study protocol in Studies 1 and 2. All subjects took their facial images using eMR Pro three times a day: during the morning after wake-up, during the morning after face wash and during the evening after face wash. Baseline data were collected from week -4 to week -1. Then, the GFF skin care formula was used from week 1 to week 4.

**Figure 4 jcm-10-02502-f004:**
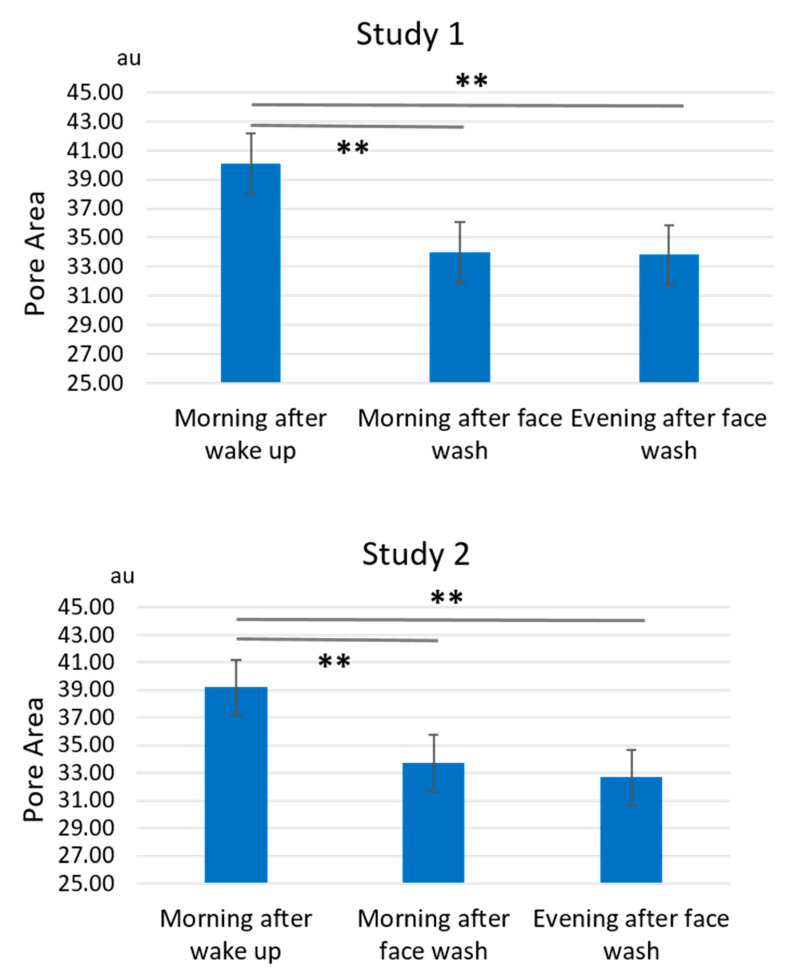
Daily fluctuation of pore area. Facial pore area is significantly larger during the morning after wake-up than during the morning after face wash or during the evening after face wash in both Study 1 and Study 2 at baseline period week -4 to week -1. ** *p* < 0.01.

**Figure 5 jcm-10-02502-f005:**
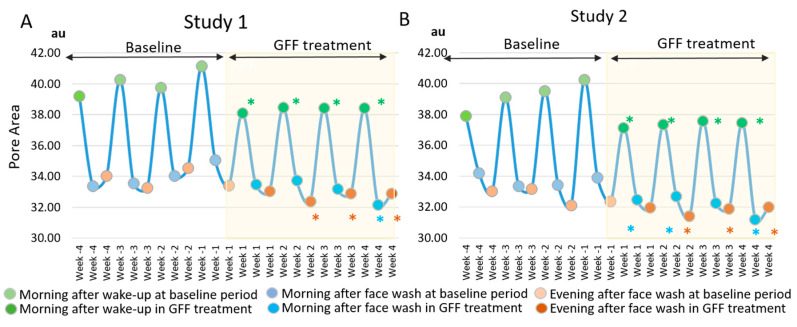
GFF treatment alleviates daily fluctuation of pore area. The “Morning after wake-up”-largest fluctuation pattern of pore area is depicted at baseline period through week -4 to week -1. GFF treatment significantly reduces the fluctuation of the pore area during week 1 to week 4. *: *p* < 0.05 compared with “morning after wake-up” at week -4”. *: *p* < 0.05 compared with “morning after face wash” at week -4”. *: *p* < 0.05 compared with “evening after face wash” at week -4”. (**A**); Study 1, (**B**); Study 2.

**Figure 6 jcm-10-02502-f006:**
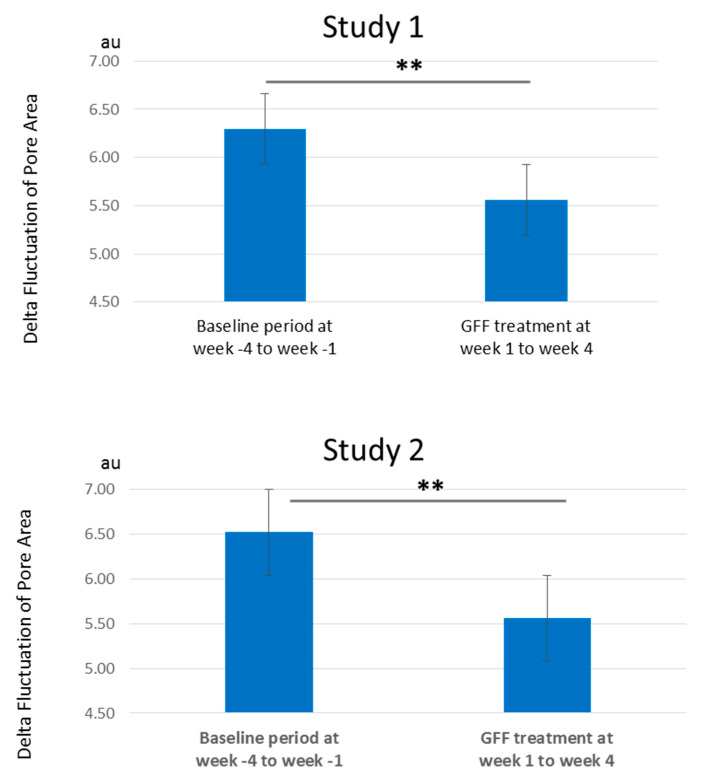
Delta fluctuation of pore area. Delta fluctuation of the pore area is significantly ameliorated by GFF treatment from week 1 to week 4 compared with baseline period from week -4 to week -1. **: *p* < 0.01.

**Figure 7 jcm-10-02502-f007:**
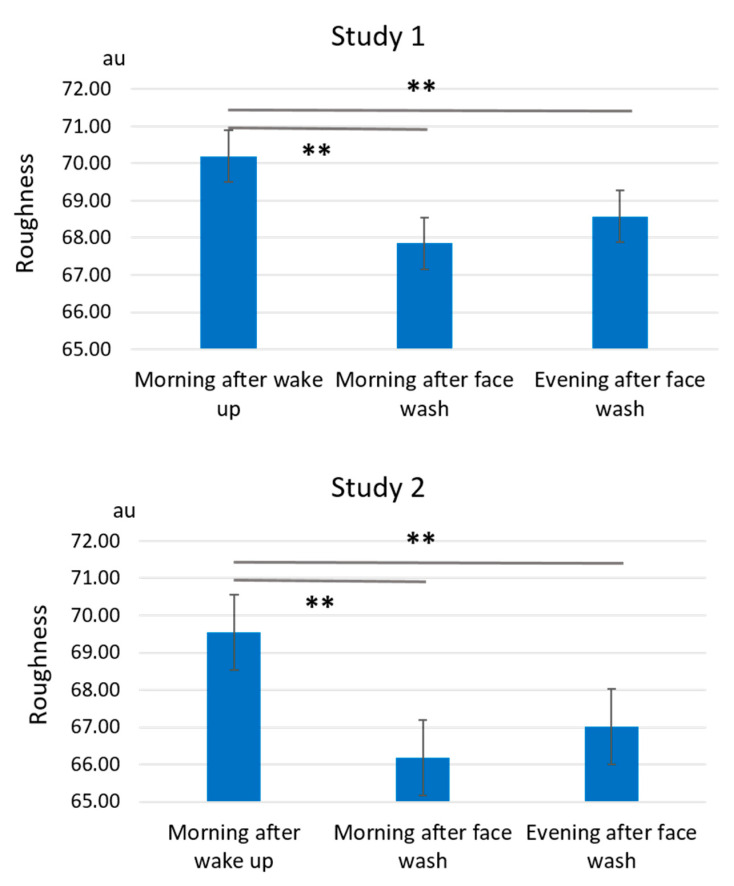
Daily fluctuation of roughness. Facial roughness is significantly larger during morning after wake-up than that during the morning after face wash or during the evening after face wash both in Studies 1 and 2 at the baseline period week -4 to week -1. **: *p* < 0.01.

**Figure 8 jcm-10-02502-f008:**
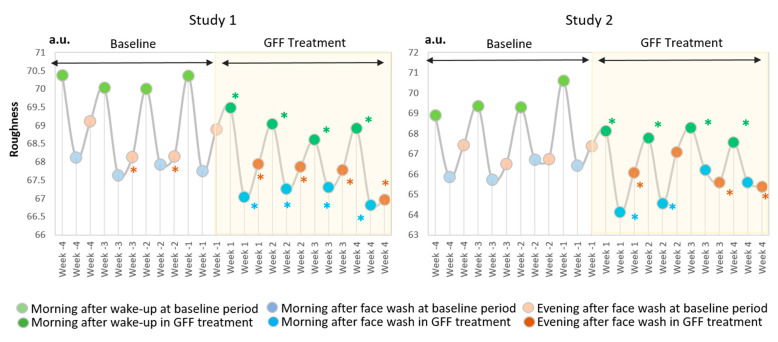
GFF treatment decreases daily fluctuation of roughness. The “Morning after wake-up”-largest fluctuation pattern of roughness is demonstrated at baseline period through week -4 to week -1. GFF treatment significantly reduces the fluctuation of roughness during week 1 to week 4. *: *p* < 0.05 compared with “morning after wake-up” at week -4”. *: *p* < 0.05 compared with “morning after face wash” at week -4”. *: *p* < 0.05 compared with “evening after face wash” at week -4”.

**Figure 9 jcm-10-02502-f009:**
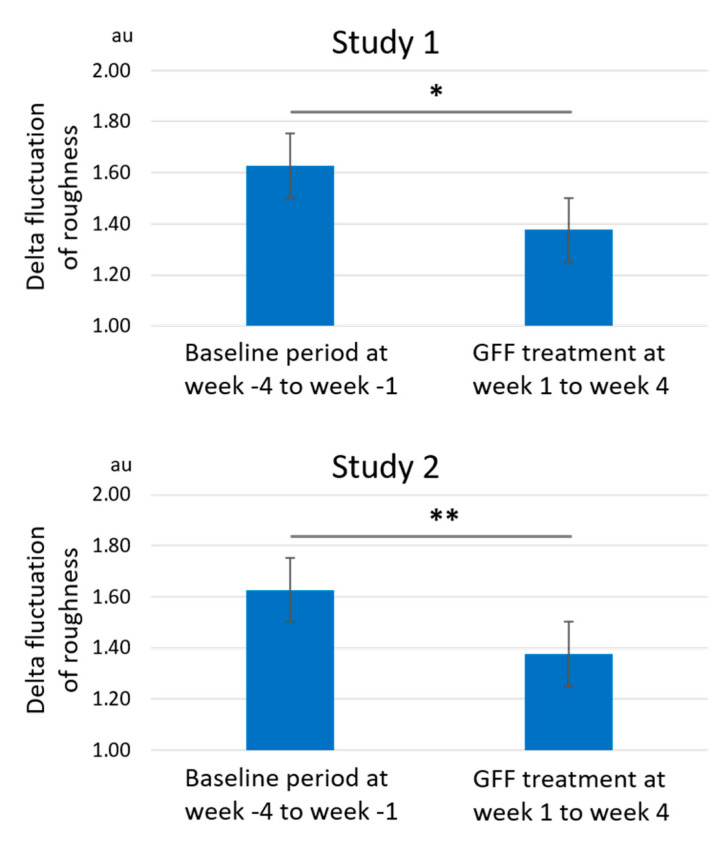
Delta fluctuation of roughness. Delta fluctuation of roughness is significantly ameliorated by GFF treatment from week 1 to week 4 compared with baseline period from week -4 to week -1. *: *p* < 0.05. **: *p* < 0.01.

**Figure 10 jcm-10-02502-f010:**
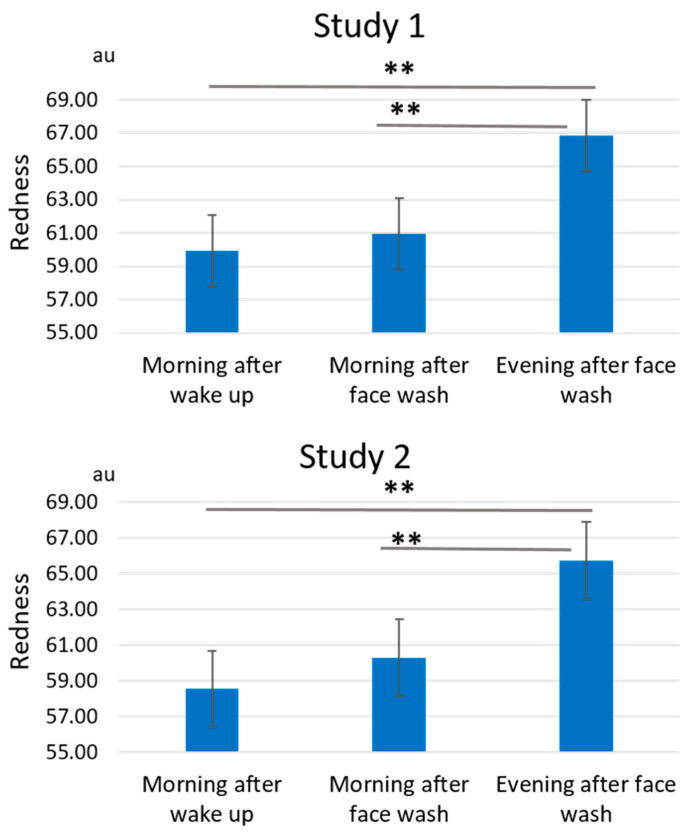
Daily fluctuation of redness. Facial redness is significantly larger during the evening after face wash than that during the morning after wake-up or during the morning after face wash in both Studies 1 and 2 at baseline period, week -4 to week -1. **: *p* < 0.01.

**Figure 11 jcm-10-02502-f011:**
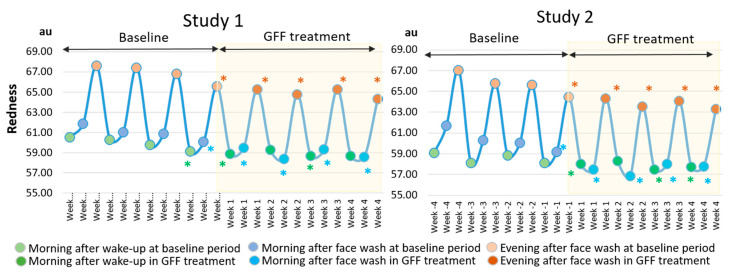
GFF treatment alleviates daily fluctuation of redness. The “evening after face wash”-largest fluctuation of redness is shown at baseline period through week -4 to week -1. GFF treatment significantly improves the fluctuation of redness during week 1 to week 4. *: *p* < 0.05 compared with “morning after wake-up” at week -4”. *: *p* < 0.05 compared with “morning after face wash” at week -4”. *: *p* < 0.05 compared with “evening after face wash” at week -4”.

**Figure 12 jcm-10-02502-f012:**
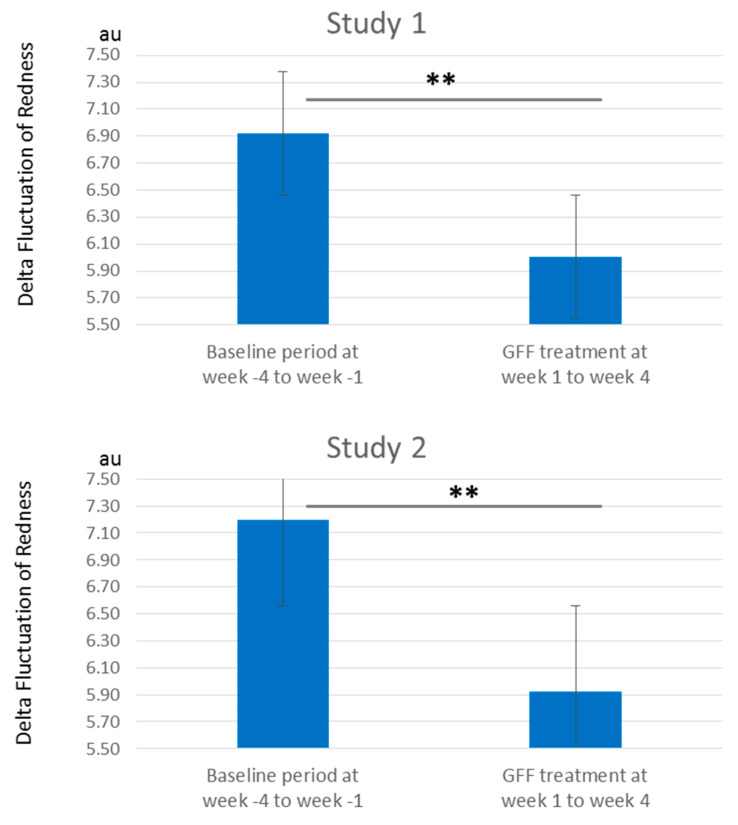
Delta fluctuation of redness. Delta fluctuation of redness is significantly decreased by GFF treatment from week 1 to week 4 compared with baseline period from week -4 to week -1. **; *p* < 0.01.

## Data Availability

The data presented in this study are available upon request from the corresponding author. The data are not publicly available because of privacy restrictions.

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
