# Peer review of "Daily Fluctuation of Facial Pore Area, Roughness and Redness among Young Japanese Women; Beneficial Effects of Galactomyces Ferment Filtrate Containing Antioxidative Skin Care Formula"

_jcm, 2021, doi:10.3390/jcm10112502_

Round 1
Reviewer 1 Report
It is a very interesting study on daily fluctuation of facial pore area. Noteworthy, the reports in the literature on this topic are scant, and this investigation is therefore welcome. The study is well structured and conducted, with a good scientific accuracy. Data are presented clearly and the conclusions are justified by the obtained results. Also, the references are adequate and updated with latest data.
Author Response
Reply to the Reviewer 1
# It is a very interesting study on daily fluctuation of facial pore area. Noteworthy, the reports in the literature on this topic are scant, and this investigation is therefore welcome. The study is well structured and conducted, with a good scientific accuracy. Data are presented clearly and the conclusions are justified by the obtained results. Also, the references are adequate and updated with latest data.
→ Thank you so much for your encouraging comments. We appreciate it very much.

Reviewer 2 Report
Very interesting and well-writing article.
- Do all authors declare that they have no conflict of interest? Please specify if any of the authors didn´t receive grants from P&G or if they are employees of the company should clarify here.
- Regarding the studies was performed just in Japanese women. “Japanese” should be added in the title.
- Regarding the two studies being conducted at different weather station, I think is important to compare the size of the pores and the redness with the weather. Study 1 was carried out between August and November (warm weather and cool temperature). Study 2 between July to Setember (very hot to pleasant hot)
- Specify who signed the ethics committee
- In the Discussion part, page 13, line 236. There is a sentence “Our preliminary experiments showed that GFF significantly decreased the sebum production of sebocytes stimulated by ultraviolet ray B irradiation in vitro (Supplementary Figure S4)”. This experiment is not explained in the Material and Methods/protocol. Maybe the authors could add a part before: 2.3 Clinical study protocol and describe it. Or if this information is already published (even on a poster in a congress), add the reference.
Author Response
Reply to the Reviewer 2
# Very interesting and well-writing article.
→ Thank you so much for your encouraging comments. We appreciate it very much.
# Do all authors declare that they have no conflict of interest? Please specify if any of the authors didn´t receive grants from P&G or if they are employees of the company should clarify here.
→ Thank you for your critical comments. According to your comments, we updated our conflicts of interest in the revise article in line 353 to 356.
# Regarding the studies was performed just in Japanese women. “Japanese” should be added in the title.
→ We agree with your comment. We added “Japanese” in the title.
# Regarding the two studies being conducted at different weather station, I think is important to compare the size of the pores and the redness with the weather. Study 1 was carried out between August and November (warm weather and cool temperature). Study 2 between July to September (very hot to pleasant hot)
→ Thank you for your helpful comments. The size of the pores and the redness were comparable between Study 1 and Study 2 as depicted in Supplementary Table S1. According to your comment, we added the following sentences in the revised article.
Line to 267 to 269
“Finally, the study 1 and study 2 were conducted at different weather situation. However, there were no significant differences in the pore area, roughness and redness between study 1 and study 2 (Supplementary Table S1).”
# Specify who signed the ethics committee
→ Thank you for your critical comments. According to your comments, we expanded our ethical statements as follows.
Line 142 to 145, Line 334 to 338.
“Ethical Statement: Data acquisition and analysis was performed in compliance with protocols approved by the Ethical Committee of Global Product Stewardship in P&G Innovation Good Kaisya (ethical approval number CT18-001 for study 1, and CT19-010 for study 2, respectively). Written informed consent was obtained from all participants prior to study placement.”
# In the Discussion part, page 13, line 236. There is a sentence “Our preliminary experiments showed that GFF significantly decreased the sebum production of sebocytes stimulated by ultraviolet ray B irradiation in vitro (Supplementary Figure S4)”. This experiment is not explained in the Material and Methods/protocol. Maybe the authors could add a part before: 2.3 Clinical study protocol and describe it. Or if this information is already published (even on a poster in a congress), add the reference.
→ Thank you very much for your critical comments. According to your comments, we added the experimental protocol in the revised article as follows.
Line 147 to 155
- Production of sebum from SEB1
SEB-1, immortalized human sebocytes, were cultured in standard medium consisting of DMEM with 5.5 mM glucose/Ham's F-12 3:1, fetal bovine serum 2.5%, adenine 1.8 × 10−4 M, hydrocortisone 0.4 μg ml−1, insulin 10 ng ml−1, epidermal growth factor 3 ng ml−1, and cholera toxin 1.2 × 10−10 M [9]. They were treated with or without ultraviolet ray B (UVB) irradiation (25mJ/cm2) in the presence or absence of 10% GFF obtained from P&G Godo Kabusikigaisha, Kobe, Japan [10]. After 24hrs incubation, sebum production of EB-1 cells was visualized using an Adipocyte Fluorescence Staining Kit (Cosmo Bio Co., Ltd., Tokyo, Japan).
Thank you so much again for your comments.
We hope the revised article is now suitable for publication in JCM.

Reviewer 3 Report
I find it a very interesting article, however, it is not clear to me what´s the main objective of the study. It is not described in the introduction, only in the abstract In materials and methods nothing is said about the formulation of the cream that is essential.
Author Response
Reply to the Reviewer 3
# I find it a very interesting article, however, it is not clear to me what´s the main objective of the study. It is not described in the introduction, only in the abstract
→ Thank you very much for your helpful comment We agree with your comment. According to your comment, we added the following sentence in the Introduction.
Line 71 to 73
These results suggest that the daily fluctuation of facial skin conditions is a potential new target field for investigating healthy skin maintenance.
# In materials and methods nothing is said about the formulation of the cream that is essential.
→ Thank you very much for your helpful comment. The GFF containing formula is Pitera® which is commercially available worldwide. We added the following sentence in the revised article.
Line 134 to 136
“GFF (Pitera®) is a yeast derived extract used as a moisturizing agent in cosmetics, and GFF skin care formula contains more than 90% GFF (Pitera®) without other skin conditioning ingredients [9].”
Thank you so much again for your comments.
We hope the revised article is now suitable for publication in JCM.
